# Novel Surgical Technique for Adolescent Idiopathic Scoliosis: Minimally Invasive Scoliosis Surgery

**DOI:** 10.3390/jcm11195847

**Published:** 2022-10-02

**Authors:** Sung Cheol Park, Sei Wook Son, Jae Hyuk Yang, Dong-Gune Chang, Seung Woo Suh, Yunjin Nam, Hong Jin Kim

**Affiliations:** 1Department of Orthopedic Surgery, Korea University Anam Hospital, College of Medicine, Korea University, Seoul 02841, Korea; 2Department of Orthopedic Surgery, Inje University Sanggye Paik Hospital, College of Medicine, Inje University, Seoul 01757, Korea; 3Department of Orthopedic Surgery, Korea University Guro Hospital, College of Medicine, Korea University, Seoul 08308, Korea

**Keywords:** adolescent idiopathic scoliosis, AIS, minimally invasive scoliosis surgery, minimally invasive spine surgery, MIS, MISS, novel technique

## Abstract

Despite advancements in instruments and surgical techniques for adolescent idiopathic scoliosis (AIS) surgery, conventional open scoliosis surgery (COSS) is usually required to achieve satisfactory deformity correction using various distinct surgical techniques, such as rod derotation, direct vertebral rotation, facetectomies, osteotomies, and decortication of the laminae. However, COSS is accompanied by significant blood loss and requires a large midline skin incision. Minimally invasive surgery (MIS) has evolved enormously in various fields of spinal surgery, including degenerative spinal diseases. MIS of the spine has some advantages over conventional surgery, such as a smaller incision, less blood loss and postoperative pain, and lower infection rates. Since the introduction of MIS for AIS in 2011, MIS has been reported to have comparable outcomes, including correction rate with some usual advantages of MIS. However, several complications, such as dislodgement of rods, wound infection, and hypertrophic scar formation, have also been reported in the initial stages of MIS for AIS. We devised a novel approach, called the coin-hole technique or minimally invasive scoliosis surgery (MISS), to minimize these complications. This article aimed to introduce a novel surgical technique for AIS and provide a preliminary analysis and up-to-date information regarding MISS.

## 1. Introduction

Minimally invasive surgery (MIS) has become a noteworthy trend in degenerative spinal disorders, including disc herniation, stenosis, and deformity [1,2]. It has several advantages over conventional open surgery, such as less blood loss and postoperative pain, paraspinal muscle sparing, lower infection rates, and smaller incisions. Subsequently, it can reduce postoperative morbidities, leading to a short length of hospital stay (LOS) and rapid return to daily life. 

The main goals of surgery for adolescent idiopathic scoliosis (AIS), the most common type of scoliosis, are correction of coronal and sagittal imbalances and maintenance of correction [3]. The application of MIS techniques in AIS has been considered challenging due to much larger curves, longer instrumentation, three-dimensional deformity, and severe vertebral rotation compared to adult scoliosis [4,5]. Additionally, it requires various distinct surgical techniques, such as rod derotation, direct vertebral rotation (DVR), facetectomies, osteotomies, and decortication of the laminae, which generally require conventional open surgery [6,7]. Nevertheless, there have been concerns regarding conventional open scoliosis surgery (COSS). In particular, a large longitudinal incision could occasionally discourage AIS patients, mainly teenagers, from undergoing surgery and could be associated with postoperative dissatisfaction. 

Sarwahi et al. first documented minimally invasive scoliosis surgery (MISS) for AIS using three skin incisions with a length of 2 inches and a freehand pedicle insertion technique combined with facetectomy and facet fusion [7]. The authors reported a similar correction rate and satisfactory fusion compared with COSS with fewer blood transfusions, postoperatively [5]. Since the introduction of MISS, there have been several reports regarding the outcomes of MISS and comparisons with COSS [4,8,9,10]. Despite comparable radiological outcomes and several inspiring results regarding LOS, scar length, and intraoperative blood loss, a few complications including wound infection, wound dehiscence resulting from implant prominence, rod dislodgement, and hypertrophic scarring have also been reported. 

This article aimed to introduce a novel MISS technique and provide up-to-date information with a perspective on MISS. This review does not present a standard MISS technique or indicate the superiority of MISS over COSS. 

## 2. Considerations in Determining MISS or COSS

The flexibility of the curve is a key factor in determining whether to perform MISS [5,11]. Although there has been controversy regarding the assessment of flexibility among the various methods, flexibility is commonly evaluated using active bending radiographs [12,13]. Additionally, a curve with an extremely large Cobb angle is not regarded as flexible and suitable for MISS [5,11]. Consequently, MISS is generally indicated for patients with a Cobb angle of less than 80° in our group. It appears to be difficult to insert pedicle screws using the MISS technique into vertebral bodies, belonging to the extremely large curve due to the accompanying rotational deformity. Additionally, all correction maneuvers should be available to achieve satisfactory correction for patients with extremely large curves. However, there is a lack of clear indications or guidelines regarding MISS in AIS patients. Therefore, it should be determined based on the Cobb angle, flexibility, and preference of patients and their parents after comprehensive counseling. 

## 3. Surgical Techniques

### 3.1. Former Technique

The first MIS for AIS, introduced by Sarwahi et al., used three 2-inch-sized midline skin incisions to approach 11–13 vertebrae under the guidance of intraoperative fluoroscopy [7]. The facet joints in the thoracic spine could be located using the tip of the spinous process of the upper vertebra because the spinous process of the thoracic spine is aligned with the level of the distal facet joint. In the lumbar spine, stab incisions on the fascia in line with muscle fibers are made over the facet joints because the facets are usually palpable. After exposing the facet joints, facetectomy was performed for easy localization of the pedicle screw entry points. After preparation of the fusion bed, a sponge soaked with bone morphogenetic protein (BMP) or other bone substitute material with local autograft was applied to the fusion bed before the pedicle screw insertion to encourage fusion. There were skipped levels without instrumentation between the separate skin incisions. The rods were inserted from the cranial to the caudal direction to use overlapping laminae of the thoracic spine as a barrier to prevent inadvertent entry into the spinal canal. Most reduction maneuvers, such as rod derotation or translation, DVR, additional compression and distraction, and in situ rod bending can be performed.

### 3.2. Our Group’s Technique

We devised our own MISS procedure, also called the coin-hole technique, to minimize the risk of complications, such as surgical site infection, hypertrophic scarring, and rod dislodgement [10,14]. The fusion levels were preoperatively determined based on Lenke’s or Suk’s classification using whole-spine standing plain radiographs [15,16]. To increase the redundancy or movability of soft tissues, the patient was positioned prone with an operating table in a hyperextended position. Incision sites and the upper and lower most vertebrae to be instrumented are checked using C-arm fluoroscopy because the precise location of the skin incision is particularly important for MIS. After marking the key anatomical landmarks, two or three midline skin incisions with a length of approximately 3 cm were made. Each incision is supposed to cover four to seven segments in the thoracic region and three to four vertebrae in the lumbar area because of skin redundancy after subcutaneous undermining dissection. 

At the level of the fascia, we used different approaches between the thoracic and lumbar areas: (1) erector spinae muscle splitting for the thoracic spine; (2) the Wiltse approach accessing the intermuscular septum for the lumbar spine. After passing through the muscle layer, the facet joints are exposed by applying a tubular retractor system or right-angled retractors, specially designed with various lengths and widths (Figure 1).

After removing the facet capsule, the entry points of the pedicle screw were marked at the lateral 1/3 of the superior articular process base. Pedicle screws were inserted into all segments without skipping any level. We used a guidewire and cannulated screws to maintain the entry point and trajectory of the pedicle screws against the interruption of the surrounding skin and muscles. We also used cannulated polyaxial long head reduction screws (GS medical Co., Ltd., Seoul, Korea) to ease rod assembly (Figure 2). After inserting the guidewire, a specially designed reamer with a cutting head (Medyssey Co., Ltd., Seoul, Korea) was inserted over the guidewire to prepare the fusion bed and release the rigidity of the curvature by damaging the ligamentous and bony structures (Figure 3). Chipped cancellous bone allografts were applied to the fusion bed before the insertion of the pedicle screw, which facilitates bone fusion by compressing the graft materials between the bone surface and pedicle screws. An additional stab incision was occasionally needed when the original midline incision was unsuitable for the screw trajectory of the upper or lower most vertebrae. A low-profile screw would be used in this case to prevent skin protrusion. The sagittal profile required for satisfactory correction was measured with three-dimensional computed tomography preoperatively. The rods were slightly overbent compared to the desired curve profile because we think that the rod contour generally becomes somewhat gentler than initially set. To properly contour the rods, we utilized the actual size plain radiograph of the whole spine, which was displayed on a portable monitor wrapped with sterilized vinyl. The rods were contoured into the desired curve profile by comparison with the actual size plain radiograph. Rods with a diameter of 6.0 mm were inserted from the cranial to the caudal direction to avoid accidental entry into the central canal. After inserting the rods, deformity correction was performed with derotation maneuver (Figure 4). More than 90% of our patients underwent thoracoplasty to resolve the residual hump after correction. Four to five ribs were usually resected with a size of 1–2 cm using a 5 mm diamond burr at the most prominent area without an additional skin incision.

In summary, the distinctive features of our technique are as follows: (1) setting the operating table in a hyperextended position (2) use of tubular or specially designed right-angled retractors with various lengths and widths, (3) use of a guide wire and cannulated instruments system to reduce radiation exposure by maintaining the pathway for the pedicle screws, (4) novel fusion technique to achieve satisfactory fusion rates without BMP by using a specially designed reamer for preparation of fusion bed and applying fusion materials prior to pedicle screw insertion to compress grafted materials, (5) all-pedicle screw fixation without skipped levels, and (6) thoracoplasty without an additional skin incision by undermining the skin.

## 4. Comparisons with Conventional Open Scoliosis Surgery Regarding Surgical Outcome

Previous studies by our group regarding MISS documented satisfactory radiological outcomes, such as the sagittal vertical axis (SVA), clavicle angle, and correction rate of the Cobb angle, compared to preoperative values [10,11]. In a comparison with COSS, the correction rate of the Cobb angle in MISS was significantly lower than that in COSS [4]. Nevertheless, radiological parameters representing global balance, such as SVA and coronal balance in the MISS group were comparable with those in the COSS group (Figure 5). 

Regarding clinical outcomes, there was no significant difference in the results of the Scoliosis Research Society-22 questionnaire, the assessment of the health-related quality of life in patients with AIS, between MISS and COSS [17]. Additionally, although not statistically significant, there was a tendency for the MISS group to have a higher satisfaction score than the COSS group.

It has been reported that MISS has advantages in terms of estimated blood loss and scar length. In contrast, the duration of surgery and anesthesia were significantly longer in patients with MISS than in those with COSS. Several complications, such as hemothorax, surgical wound infection, and wound dehiscence, have been reported after MISS. However, after we started to perform thoracoplasty using a diamond burr with a Cobb elevator applied under the rib to protect the surrounding tissue rather than a Kerrison rongeur, no patient had hemothorax after surgery. There were no identified patients with mechanical and instrument-related complications, including proximal junctional kyphosis or failure and rod breakage, except for a patient with screw loosening. Table 1 shows a group of patients who underwent MISS for AIS between March 2015 and February 2020 in our hospital.

## 5. Discussion

Although MISS has been reported to have several advantages regarding blood loss, LOS, and scar length, it has also been suggested to be technically challenging and to increase operation time and radiation exposure for both surgeons and patients [4,5,8,10]. Nevertheless, several surgeons have been trying to apply MIS techniques for the correction of AIS, being eager to reduce surgical morbidities and bring cosmetic benefits. Moreover, efforts to reduce the surgical burden have extended to other types of deformities, such as neuromuscular scoliosis [18,19]. 

It may be difficult to achieve satisfactory arthrodesis in MIS settings [7]. Iliac crest autograft and BMP could be considered as available options to increase fusion potential in various spinal fusion surgeries [20,21]. Although autologous bone is considered the gold standard bone graft material, it is well known to be associated with several graft site morbidities, such as pain, hematoma formation, infection, and pelvic fracture [22,23]. BMP has been widely utilized to increase fusion potential in various spinal fusion surgeries [24,25]. Although several previous studies have reported no significant difference in acute complications and malignancy risk between pediatric patients treated with and without BMP, complications such as wound infections, seromas, and heterotopic bone development have also been reported [26,27,28,29,30]. Moreover, its long-term effects on child-bearing age remain unknown [7]. Meanwhile, we obtained favorable fusion rates without BMP by the following sequential novel techniques: (1) using a specially designed reamer to secure a good fusion bed; (2) applying fusion materials on the fusion bed before the insertion of pedicle screws; (3) compressing these materials by inserting screws [11]. 

There has been a controversy with regard to the correction potential of MISS. While some authors have suggested that MISS could achieve radiological outcomes comparable to COSS, others have reported relatively lower correction rates [5,8]. Although our previous study reported a lower correction rate for the Cobb angle of the main curve in MISS, global alignment parameters were not significantly different between MISS and COSS [4]. Although most correction maneuvers can be performed theoretically with MISS, it is the case that some reduction methods are inconvenient to perform as in COSS due to limited surgical field and various obstacles, such as surrounding paraspinal muscles and ligaments [5,11]. During the process of pedicle screw insertion in our novel MISS, a specially devised reamer with a cutting head was used as described, resulting in damage to the stabilization structures, such as bony structures and ligaments, and subsequent reduction in rigidity of the deformed spine. We assume that this procedure can improve the correction potential of MISS. 

MISS has been known to be associated with extensive radiation exposure, particularly detrimental to children or adolescents [31,32]. Radiation exposure is inevitable during MISS because there are difficulties in finding the entry point and trajectory of pedicle screws and maintaining this trajectory consistently because of the limited view and interference of surrounding tissues. Therefore, we used cannulated screws with a guidewire and free-hand techniques. A tubular retractor system or right-angled retractors of various lengths and widths, and a fiber-optic light source are used to secure adequate exposure to easily find the entry point. Additionally, the sequential process of screw insertion can be performed without serial radiographic checks by using a guidewire and specially designed cannulated instruments.

Several complications associated with surgical wounds have been reported. Sarwahi et al. described a patient with hypertrophic scarring after MISS due to the possible continuous retraction of the incision [5]. This could be closely associated with patient dissatisfaction after surgery because the cosmetic outcome is one of the major concerns of adolescent patients [11]. Additionally, other wound problems, such as deep infection, wound breakdown, and draining sinus, have been documented [5,8]. Several adjustments have been made to other reported techniques to minimize soft tissue damage during surgery. First, the operating table was set to a hyperextended position to secure redundancy of the back skin. The incision length can also be reduced by increasing the skin redundancy. Second, we used Wiltse’s approach using the intermuscular cleavage plane to preserve the paraspinal muscles through a midline skin incision after undermining the layer between the fascia and subcutaneous fat for the thoracolumbar and lumbar area. Third, we utilized tubular retractors to minimize soft tissue manipulation compared with manual retractors. However, wound complications could not be completely avoided in our group [4]. Further adjustment of the procedure and/or development of instruments are needed.

Despite constant efforts and advancements in surgical techniques and instruments, there are still a few disadvantages of MISS. In particular, a much longer operation time compared with COSS has been consistently suggested [4,5,8]. This might be due to several additional procedures and technical challenges, including pedicle screw insertion and rod assembly. However, since this could also be a reflection of the learning curve, operation time is expected to decrease as surgeons gain more experience. Similarly, the reported complications are also likely to decrease in the future. 

This study had some limitations. First, previously reported outcomes could change in the future with the possibility of improvement. The nature of the learning curve should be considered when evaluating the effectiveness or weakness of MISS. Second, this review article focused on our technical know-how and general concepts of MISS and presented the outcomes of a part of patients. Further systematic study to analyze the outcomes of MISS depending on the accumulation of experience should be conducted in the future. Third, we did not demonstrate outcomes depending on curve type or flexibility. Therefore, it seems unreasonable to suggest MISS as a standard method for AIS. More experience with surgery for various curve types is required. Finally, this is not a systematic review with meta-analysis, but a description of our MIS technique for AIS. Indeed, several previous meta-analyses included only a few studies that were mainly retrospective with a small number of patients and a relatively short-term follow-up period. There is still a paucity of large-scale studies or randomized controlled trials on MIS for pediatric spinal deformities. Well-controlled studies are required to confirm the efficacy of MISS. Despite these limitations, this article thoroughly describes our novel MISS technique in comparison with other reported techniques and suggests several remaining problems to solve. 

## 6. Conclusions

MISS could be a reasonable option for AIS, providing comparable deformity correction and fusion with several advantages, such as low surgical morbidities and cosmetic benefits. With advances in surgical techniques and instruments, MISS is likely to become one of the main concepts for the management of pediatric spinal deformities in the future.

## Figures and Tables

**Figure 1 jcm-11-05847-f001:**
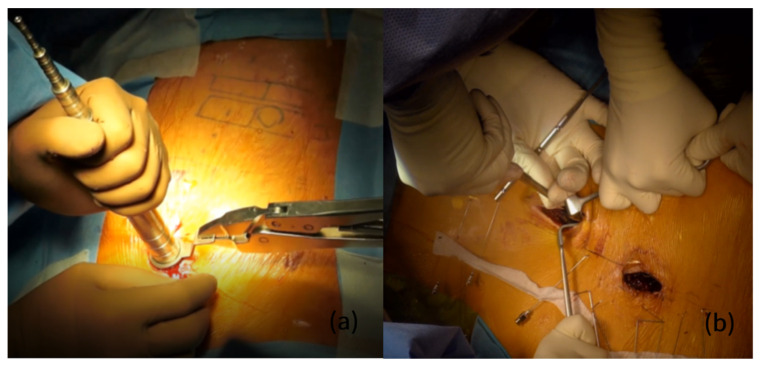
(**a**) Placement of tubular retractor to expose the targeted facet joints. (**b**) Exposing the surgical field using right-angled retractors with various lengths and widths.

**Figure 2 jcm-11-05847-f002:**
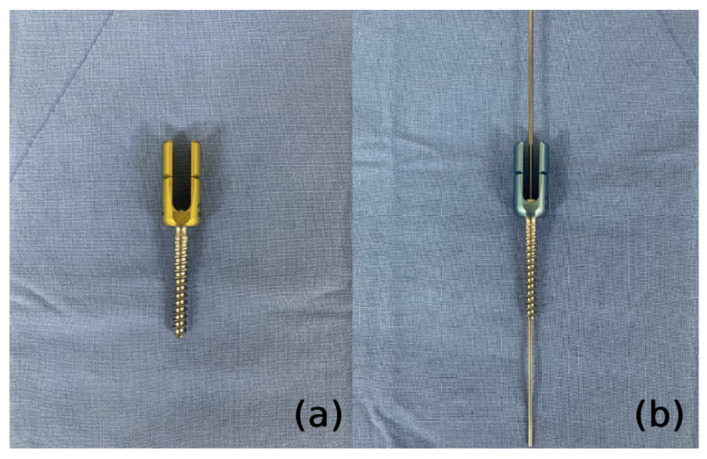
(**a**) A polyaxial long head reduction screw to ease rod assembly. (**b**) A cannulated screw inserted over the guidewire.

**Figure 3 jcm-11-05847-f003:**
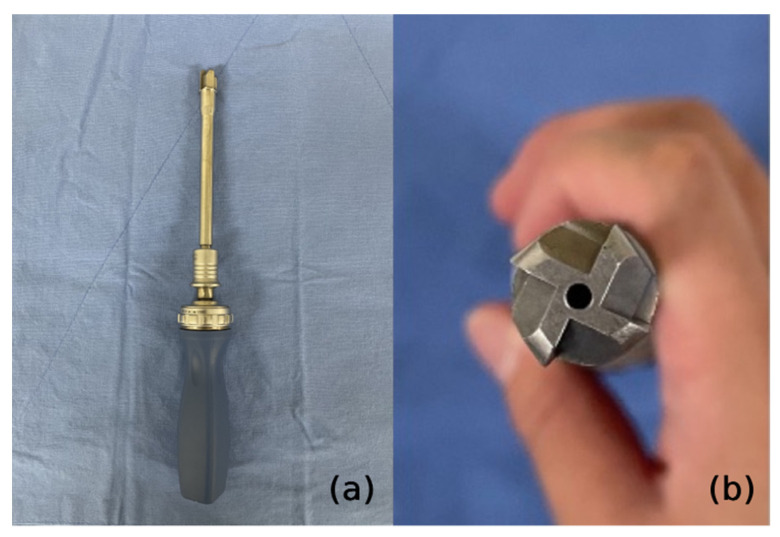
(**a**) A specially designed cannulated reamer (**b**) A cutting head to prepare the fusion bed and damage the surrounding ligamentous and bony structures.

**Figure 4 jcm-11-05847-f004:**
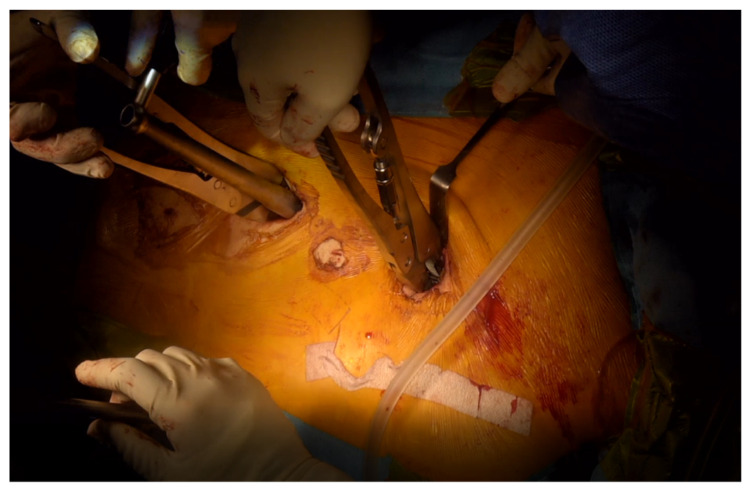
Deformity correction technique with derotation maneuver.

**Figure 5 jcm-11-05847-f005:**
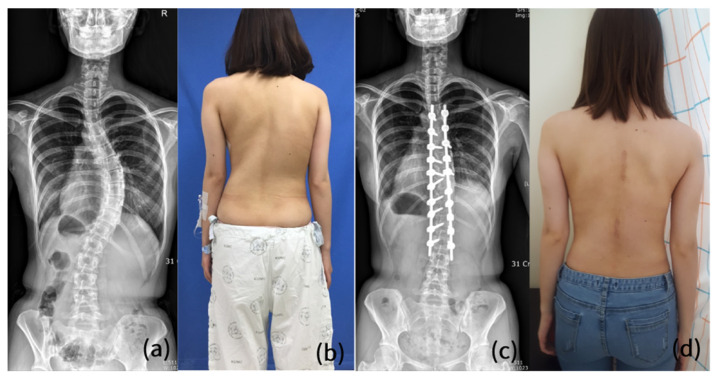
A 16-year-old female AIS patient who underwent MISS with two skin incisions. (**a**,**b**) An AIS patient with Lenke type 1 curve and Risser’s stage 5. (**c**,**d**) The postoperative whole spine anteroposterior plain radiograph and clinical photo showing satisfactory coronal balance. AIS, adolescent idiopathic scoliosis; MISS, minimally invasive scoliosis surgery.

**Table 1 jcm-11-05847-t001:** Baseline characteristics and outcomes of the patients who underwent minimally invasive scoliosis surgery.

	Values
Total number of patients	52
Age	15.31 ± 2.03 *
Sex (*n*): men/women	2/50
Body mass index (kg/m^2^)	19.27 ± 2.79 *
Lenke classification (*n*): 1/2/3/4/5/6	39/4/6/0/2/1
King classification (*n*): 1/2/3/4/5	7/11/26/5/3
Duration of surgery (min)	390.00 ± 91.69 *
Estimated blood loss (mL)	1135.96 ± 678.20 *
Number of levels operated (*n*)	10.88 ± 1.37 *
Radiological parameters	
Preoperative Cobb angle (°)	62.00 ± 9.75 *
Cobb angle in bending films (°)	46.13 ± 14.33 *
Flexibility (%)	26.86 ± 15.75 *
Postoperative Cobb angle (°)	22.40 ± 7.03 *
∆ Cobb angle (°)	38.02 ± 8.70 *
Correction rate (%)	64.02 ± 8.96 *

* Mean ± Standard deviation.

## Data Availability

The data presented in this study are available on reasonable request from the corresponding author.

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
