# Peer review of "Novel Surgical Technique for Adolescent Idiopathic Scoliosis: Minimally Invasive Scoliosis Surgery"

_jcm, 2022, doi:10.3390/jcm11195847_

Round 1
Reviewer 1 Report
The authors present work on the possibility of using MISS for AIS.
The idea is certainly interesting as is the technique described.
The problem is that it is not apparent from the work how many patients have been selected and operated on with the technique the authors propose and describe.
If it is only one patient - the one shown in the photo of the work - unfortunately it means nothing. One should then talk about a case report or technical note, otherwise they should still present a small case report of 5-10 patients.
I would suggest reshaping the title and include a chapter on the limitations of the study and technique themselves, or give data on multiple patients even then presenting only one.
Reviewer 2 Report
I enjoyed reading about the progression of the MIS approach for surgical treatment of AIS by instrumented spinal fusion. However, the paper is limited to a description of their surgical approach. The manuscript lack data about the case series such as patient demographics, mean Cobb angle pre and post op, surgical time, number of cases performed, and experience of the operative surgical team.
IF the editors decide to accept the manuscript as a pure technique article then the authors should add more photos of the technique to show screw , rod, and tube placement placement.
Reviewer 3 Report
The authors present a readily comprehensible presentation of a minimally invasive surgical technique for adolescent scoliosis. The review is very well written and the discussion addresses the essential aspects of this technique with the associated issues.
In my view, there are only a few minor comments that should be addressed.
1. The authors refer several times to the first published minimally invasive surgical techniques for AIS - overall, however, after reading the article it is difficult to understand where exactly the innovative differences to the previously published techniques lie and what makes the presented technique so new. Is it the use of the tubular retractor system? The unique approach to securing a spondylodesis? Here, a small paragraph summarising briefly and concisely the innovations offered by the presented technique (for example, as a paragraph in results) would be helpful.
2. Even if this paper is only intended to present the surgical technique. For a surgical technique of this kind, it is essential for the reader to know how often the authors have performed this surgical technique. It makes a big difference whether the technique was tested and established for the first time (only on one patient) or on a whole collective of patients. As far as I understand, the authors have used this technique on a group of patients. A concise tabular presentation should be added. In the context of this publication, this does not have to be a statistical analysis; a descriptive overview of the operations performed so far is sufficient, ideally with blood loss, operation duration and complications.
3. Even if the description of the surgical technique in the text is very detailed and easy to follow. Describing a surgical technique using illustrations for a schematic representation is always easier to understand and prevents misinterpretations - authors should consider this as a matter of principle.
Points 1 and 2 in particular should still be addressed before publication.
Point 3 is a matter of principle and represents a personal recommendation on my part.
Round 2
Reviewer 2 Report
PLease include the number of patients in your cohort used to generate Table 1.
Author Response
We appreciate your kind comment to improve the quality of our study. We have added the total number of patients in this cohort in the Table 1.